**Data Availability Statement:** The whole entire data set is controlled and regulated by the Department of Veteran Affairs. Prior approval to release the data

# Assessment of mitochondrial respiratory capacity using minimally invasive and noninvasive techniques in persons with spinal cord injury

Raymond E. Lai[1,2], Matthew E. Holman[1,3], Qun Chen[4], Jeannie Rivers[5], Edward J. Lesnefsky[4,6], Ashraf S. Gorgey[1,2]*

1 Spinal Cord Injury and Disorders, Hunter Holmes McGuire VA Medical Center, Richmond, Virginia, United States of America, 2 Physical Medicine and Rehabilitation, Virginia Commonwealth University, Richmond, Virginia, United States of America, 3 Physical Therapy, Virginia Commonwealth University, Richmond, VA, United States of America, 4 Division of Cardiology, Division of Internal Medicine, Pauley Heart Center, Department of Medicine, Virginia Commonwealth University, Richmond, VA, United States of America, 5 Surgical Service, Hunter Holmes McGuire VA Medical Center, Richmond, VA, United States of America, 6 Medical Service, Hunter Holmes McGuire VA Medical Center, Richmond, VA, United States of America

* ashraf.gorgey@va.gov

## Abstract

### Purpose

Muscle biopsies are the gold standard to assess mitochondrial respiration; however, biopsies are not always a feasible approach in persons with spinal cord injury (SCI). Peripheral blood mononuclear cells (PBMCs) and near-infrared spectroscopy (NIRS) may alternatively be predictive of mitochondrial respiration. The purpose of the study was to evaluate whether mitochondrial respiration of PBMCs and NIRS are predictive of respiration of permeabilized muscle fibers after SCI.

### Methods

Twenty-two individuals with chronic complete and incomplete motor SCI between 18–65 years old were recruited to participate in the current trial. Using high-resolution respirometry, mitochondrial respiratory capacity was measured for PBMCs and muscle fibers of the vastus lateralis oxidizing complex I, II, and IV substrates. NIRS was used to assess mitochondrial capacity of the vastus lateralis with serial cuff occlusions and electrical stimulation.

### Results

Positive relationships were observed between PBMC and permeabilized muscle fibers for mitochondrial complex IV (r = 0.86, *P* < 0.0001). Bland-Altman displayed agreement for complex IV (MD = 0.18, LOA = -0.86 to 1.21), between PBMCs and permeabilized muscles fibers. No significant relationships were observed between NIRS mitochondrial capacity and respiration in permeabilized muscle fibers.

is required. The raw data supporting the conclusions for this manuscript will be made available by the corresponding author, after approval from our research department by the ACOS of research to any qualified researcher. Data are available from the Department of Veteran affairs (contact via angela.davis@va.gov) for researchers who meet the criteria for access to confidential data.

**Funding:** AG: Ashraf Gorgey This study was supported by the DoD-CDRMP (W81XWH-14-SCIRP-CTA). The funders had no role in study design, data collection and analysis, decision to publish, or preparation of the manuscript.

**Competing interests:** The authors have declared that no competing interests exist.

**Abbreviations:** ADP, Adenosine diphosphate; AIS, American Spinal Injury Association impairment scale; ATP, Adenosine triphosphate; BMI, Body mass index; BMR, Body metabolic rate; BP, Blood pressure; Hba1c, Glycosylated hemoglobin; HHb, Deoxygenated hemoglobin; NIRS, Near-infrared spectroscopy; NMES, Neuromuscular electrostimulation; PBMC, Peripheral blood mononuclear cells; SCI, Spinal cord injury; T2DM, Type 2 diabetes mellitus; VL, Vastus lateralis; $VO_2$, Maximum oxygen uptake.

## Conclusions

This is the first study to explore and support the agreement of less invasive clinical techniques for assessing mitochondrial respiratory capacity in individuals with SCI. The findings will assist in the application of PBMCs as a viable alternative for assessing mitochondrial health in persons with SCI in future clinical studies.

## Introduction

Within most eukaryotic cells, mitochondria serve a pivotal role as a robust ATP-generating system [1, 2]. Dysregulation of mitochondria's dynamic process of biogenesis, remodeling, and degradation, result in unfavorable physiological outcomes including diminished energy production, increased reactive oxygen species, and even cell death [3]. Mitochondrial dysfunction may contribute to obesity and insulin resistance which can lead to the development of cardiovascular disease and T2DM, chronic diseases that are commonplace within the spinal cord injury (SCI) population [4, 5]. As such, methods for evaluating mitochondrial function hold critical importance during the initial stages of diagnosis as well as determining patient prognosis. Knowledge regarding changes in mitochondrial activity following SCI is limited. SCI is accompanied by cellular, metabolic, and body composition changes [6, 7]. An increase in adipose tissue combined with inactivity and skeletal muscle atrophy decreases daily energy expenditure and contributes to the obesity commonly observed in the SCI population [6, 8]. Furthermore, previous work suggests that mitochondrial mass and activity are decreased in older individuals with SCI when compared to those who are younger [9]. A previous study demonstrated that during an acute bout of electrical stimulation, persons with SCI heavily rely on carbohydrate utilization compared to fat utilization as demonstrated by the respiratory exchange ratio, which may indirectly suggest diminished mitochondrial activity [10]. However, it is still unclear whether mitochondrial dysfunction contributes to this observed pattern of substrate utilization after SCI. Multiple studies have shown that electrical stimulation training of muscle may yield increased muscle endurance, function increases in oxidative enzyme activity and gene expression [11–14]. These results suggest that improved muscle function may be mediated by mitochondrial capacity and highlights the need for clinical evaluation tools to assist in monitoring skeletal muscle function in this population [15]. An improved understanding of the cellular response of skeletal muscles in chronic SCI may help clinicians better formulate therapeutic and rehabilitative regimens for improving the long-term health and quality of life for persons living with SCI [13, 16].

Sources of mitochondrial dysfunction identified through respirometry include impaired mitochondrial membrane transport, substrate utilization issues, and disrupted fatty acid metabolism [17]. Prior studies have established protocols measuring mitochondrial activity from as little as 2–4 mg of skeletal muscle tissue [18–20]; however, these methods have not been extensively utilized in studies involving skeletal muscle biopsies from individuals with SCI. This invasive procedure requires a skilled team and technical expertise not available in most clinical settings. The use of muscle biopsies, although very helpful in studying mitochondria, is not a feasible strategy in clinical trials with large sample sizes as individuals may not consent to this procedure. Furthermore, many individuals may be concerned about the discomfort or complications associated with this invasive procedure. Less or noninvasive methods can also be used to quantify mitochondrial activity such as the measurement of blood cell mitochondrial function and skeletal muscle oxygenation [19, 21]. However, it remains

unknown whether these procedures are representative of skeletal muscle mitochondrial function after SCI.

Recent research suggests that mitochondrial activity in blood cells such as peripheral blood mononuclear cells (PBMCs) may be representative of overall mitochondrial health [19, 22–26]. PBMCs are a subpopulation of white blood cells and include mostly lymphocytes and monocytes. Because of their exposure to inflammatory signals released from skeletal muscle in times of metabolic stress, these cells may be predictive of metabolic health [27–29]. Aging individuals, as well as those with T2DM or chronic kidney disease have dysfunctional white blood cell mitochondria similar to that seen in other tissues [22, 23]. Additionally, PBMC mitochondrial activity has been shown to be related to skeletal muscle mitochondrial activity in older adults [24]. A recent *in vivo* study performed using non-human primates demonstrated that blood based cellular respirometry is significantly correlated with bioenergetic measurements generated by skeletal and cardiac muscle [19]. Despite the relative promise of these earlier studies, it is still unknown if PBMC mitochondrial function is associated with oxygen consumption from skeletal muscle biopsies in humans with SCI.

A noninvasive tool helpful in measuring mitochondrial function is near-infrared spectroscopy (NIRS) [30]. NIRS measures tissue oxygenation through the oxygen-dependent absorption of near-infrared light by hemoglobin and myoglobin [30]. This technology was combined with methodologies originally adapted for use with magnetic resonance spectroscopy to assess the oxidative capacity of skeletal muscle following brief exercise [31–35]. Compared to magnetic resonance spectroscopy, the NIRS technique is more convenient and relatively inexpensive [30]. This NIRS serial arterial occlusion technique has subsequently been validated in comparison with magnetic resonance spectroscopy [31–35] as well as invasive skeletal muscle biopsy techniques [20]. Within SCI [12, 36] and other clinical populations [37, 38], this approach has been successfully implemented; however, these examples are limited and have not been validated for specific mitochondrial complexes following SCI.

The present study seeks to evaluate whether mitochondrial respiration of PBMCs and skeletal muscle oxygenation measured noninvasively by NIRS are predictive of permeabilized muscle fiber respiration in individuals with chronic SCI. We hypothesized that mitochondrial respiratory capacity measured by both high-resolution respirometry of PBMCs and NIRS would be correlated with those observed in high-resolution respirometry of permeabilized skeletal muscle fibers. The focus is on using these less-invasive techniques to measure cellular activity with the hope of decreasing the need for invasive techniques such as muscle biopsies.

## Material and methods

### Ethical approval

All aspects of this study were approved by the McGuire VA Medical Center institutional review board and conducted in the McGuire VA Medical Center. All procedures were conducted in accordance with the ethical standards of the Helsinki Declaration of 1964 and its later amendments. Participants provided written informed consent as part of an ongoing clinical trial, registered at clinicaltrials.gov (NCT# NCT02660073). Data presented in this study are cross-sectional prior to conducting any intervention.

### Participants

Individuals with chronic SCI ($\geq$ 1-year post injury), aged between 18–65 years, characterized by low levels of physical activity, and with a body mass index (BMI) $\leq$ 30 kg/m$^2$ were recruited into the study. BMI was measured as each subject's weight divided by their height squared [39]. All were asked to read and sign a consent form approved by local ethics committees.

Participants were classified using the American Spinal Injury Association impairment scale (AIS) as either motor complete (A) or incomplete (B or C) as determined through a comprehensive physical examination by a board-certified SCI physiatrist. Subjects were excluded if they presented with a current or prior history of any of the following: cardiovascular disease (*e. g.*, myocardial infarction, heart failure, etc.), type 1 diabetes, non-optimally treated T2DM (HbA1c > 5%), uncontrolled hypertension (BP > 130/80 mmHg), and/or insulin dependence.

## Study design

The current study is based on data gathered at baseline prior to intervention from an ongoing clinical trial. Following an overnight fast (10–12 hours), participants came into the lab, voided their bladder, then underwent a muscle biopsy procedure in the early afternoon. Participants returned to the lab 3–4 days later following another overnight fast and to have a small sample of blood collected for the PBMCs isolation. Shortly that same morning, subjects began NIRS testing. In addition to the overnight fast, participants were asked to abstain from any level of physical activity for 2–3 day prior to coming on site.

## Skeletal muscle biopsy & tissue preparation

Muscle biopsy samples were obtained from all subjects in the morning following an overnight fast. Briefly, muscle biopsy specimens were collected from the right vastus lateralis (VL) using a 14-gauge Tru-Cut needle (Merit Medical Systems, South Jordan, UT, USA) under local anesthesia (2% lidocaine). Immediately following the biopsy procedure, ~20 mg was placed in ice cold biopsy preserving solution (BioPS media, 2.77 mM $CaK_2EGTA$, 7.23 mM $K_2EGTA$, 5.77 mM $Na_2ATP$, 6.56 mM $MgCl_2$ 6 $H_2O$, 20 mM taurine, 15 mM $Na_2$Phosphocreatine, 20 mM imidazole, 0.5 mM dithiothreitol, 50 mM MES hydrate, pH 7.1) for high-resolution respirometry. Muscle fibers were separated along the longitudinal axis using needle-tipped forceps under magnification yielding between 1–10 mg (average 4.9 ± 0.5 mg) of viable tissue. Due the presence of fat infiltration that is associated in individuals with SCI, an excess amount of muscle was utilized [7]. The plasma membrane of muscle fibers was permeabilized by gentle agitation for 20 min at 4°C in BioPS containing 50 µg/ml saponin [40] followed by two 4-min washes in mitochondrial respiration buffer (miR05, 0.5 mM EGTA, 3 mM $MgCl_2$ $6H_2O$, 60 mM lactobionic acid, 20 mM taurine, 10 mM $KH_2PO_4$, 20 mM HEPES, 110 mM D–sucrose, 1g/L bovine serum albumin, pH 7.1) [13].

## High resolution respirometry measurement of permeabilized muscle fibers

High-resolution $O_2$ consumption measurements of separated muscle fibers suspended in 2 ml of miR05 buffer were measured at 37°C using the OROBOROS Oxygraph-2k (Oroboros, Innsbruck, Austria). Oxygen concentration and flux were recorded with DatLab software (Oroboros, Innsbruck, Austria). Respiration was measured using the following protocol: 10 mM glutamate + 5 mM malate (complex I substrates), followed by sequential additions of 0.1 and 1 mM ADP, 10 µM cytochrome c (to test for membrane integrity), 1 µM rotenone (complex I inhibitor), 10 mM succinate (complex II substrate), 40 µM 2-thenoyltrifluoroacetone (TTFA) (complex II inhibitor), 0.5 mM tetramethyl-p-phenylenediamine (TMPD)– 10 mM sodium ascorbate (complex IV substrates), and 10 mM sodium azide (complex IV inhibitor); Oxygen flux was expressed as $pmol \cdot sec^{-1}$ normalized to mg weight of the fiber bundle [13]. Mitochondrial respiration rates were determined after correcting for non-mitochondrial oxygen consumption following the addition of the corresponding inhibitor for complex I, II, and IV. Respiration rates were expressed as inhibitor-sensitive rates to eliminate the contribution of

**(A)**

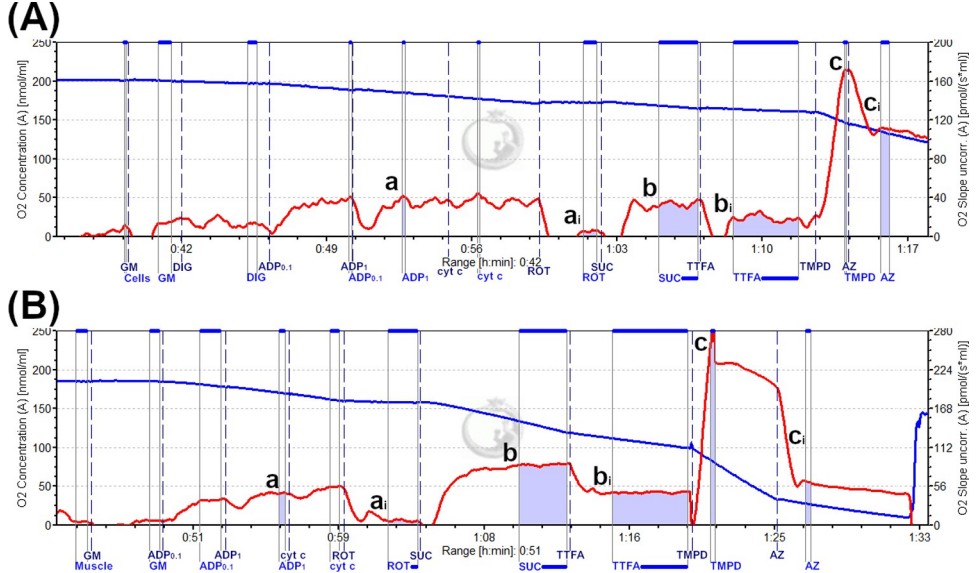

**(B)**

**Fig 1. Representative bioenergetic profile for peripheral blood mononuclear cells and permeabilized skeletal muscle fibers from one individual.** The rate of oxidative phosphorylation for PBMCs **(A)** and permeabilized skeletal muscle fiber **(B)** was measured in a high-resolution respirometer OROBOROS Oxygraph-2k. The blue line represents oxygen concentration in the oxygen chamber, and the red line represents the rate of oxygen consumption. 5.2 mg of skeletal muscle and $5.8 \times 10^6$ cells are represented in (A) and (B), respectively. Boxed blue lines indicate recorded data points. Glutamate + malate (GM) + 1mM ADP ($ADP_1$), succinate, and tetramethyl-$p$-phenylenediamine + ascorbate (TMPD) were substrates used for complex I **(a)**, II **(b)**, and IV **(c)** substrates, respectively. Rotenone (ROT), thenoyltrifluoroacetone (TTFA), azide (AZ) were used to inhibit complex I **($a_i$)**, II **($b_i$)**, and IV **($c_i$)**, respectively. Respiration rates were expressed as inhibitor-sensitive rates (*e.g.* a—$a_i$, b—$b_i$, c—$c_i$). 0.1 mM ADP ($ADP_{0.1}$) and 1mM ADP ($ADP_1$) were used to stimulate oxygen consumption. The final rate of oxidative phosphorylation pmol·$sec^{-1}$ was normalized per wet weight of skeletal muscle (mg) or $1 \times 10^6$ cells.

oxygen consumption not related to oxidation of the specific substrate [41]. A representative bioenergetic profile for skeletal muscle is shown in Fig 1.

## Blood collection and mononuclear cell isolation

Vacutainers (BD Biosciences, San Jose, CA, USA) containing ethylenediaminetetraacetic acid (EDTA) were used to collect 20 ml of blood from patients following an overnight fast ~5 days apart from their muscle biopsy procedure. Blood and muscle were collected on different days due to the inherent time-intensive efforts involved with analysis for each sample. Since the study was cross-sectional, the effects of the time gap in between sample collections should be negligible. Blood was immediately diluted with an equal amount of phosphate-buffered saline (PBS), then layered over a Ficoll-Paque (Millipore-Sigma, Burlington, MA, USA) density gradient, and centrifuged at 750 x g for 30 min in room temperature. Mononuclear cells were isolated from the interface between the plasma and the Ficoll-Paque. Collection of mononuclear cells was performed via centrifugation at 350 x g for 10 min at 4˚C, and washed twice with PBS.

## High resolution respirometry measurement of mononuclear cells

High-resolution $O_2$ consumption measurements of mononuclear cells were conducted using the OROBOROS Oxygraph-2k, and recorded by DatLab 4 software (OROBOROS Instruments, Innsbruck, Austria). Mononuclear cell pellets were resuspended in miR05 respiration buffer, then counted using the Countess II automated cell counter (Invitrogen, Carlsbad, CA,

USA). 2 ml of suspension containing intact mononuclear cells in miR05 buffer (approximately $1 \times 10^6$–$9 \times 10^6$ cells/ml) was added to the instrument's chamber. A substrate-uncoupling inhibitor protocol was used as follows: 10 mM glutamate + 5 mM malate, followed by sequential additions of 6μg digitonin (detergent) per $1 \times 10^6$ cells (final concentration: ~10–50 μg/ml), 0.1–1 mM ADP, 10 μM cytochrome $c$, 1 μM rotenone, 10 mM succinate, 40 μM 2-thenoyltrifluoroacetone (TTFA), 0.5 mM tetramethyl-p-phenylenediamine (TMPD)– 10 mM and sodium ascorbate, 10 mM sodium azide; Oxygen flux was expressed as pmol·sec$^{-1}$ normalized to $1 \times 10^6$ cells [40]. Respiration rates were corrected for non-mitochondrial respiration by using the oxygen consumption rates determined following the addition of the corresponding inhibitor for complex I, II, and IV. Similarly to skeletal muscle analysis, respiration rates for PBMCs were expressed as inhibitor-sensitive rates to eliminate the contribution of oxygen consumption not related to oxidation of the specific substrate [41]. A representative bioenergetic profile for PBMCs is shown in Fig 1.

## Near-infrared spectroscopy measurement of mitochondrial capacity

The recovery rate of deoxygenated hemoglobin (HHb), or mitochondrial capacity, was obtained with continuous wave NIRS during a serial arterial occlusion method similar to another study which demonstrated its safety for individuals with SCI [36]. A portable NIRS device with three fixed optode distances (PortaMon, Artinis Medical Systems, Elst, Netherlands) was originally employed for testing (n = 12), while a single channel device with flexible optode distances (OxyMon, Artinis Medical Systems, Elst, Netherlands) was later used for the remaining participants (n = 6). Each tool was placed longitudinally over the right VL (~10 cm above the patella) to reduce inter-limb variability with the muscle biopsy technique. Two 8 x 10 cm$^2$ surface electrodes (Uni-Patch, Wabasha, MI, USA) were also laterally seated superior to the right patella (~2–3 cm and ~30 cm respectively). Additionally, a rapid-inflating vascular cuff (Hokanson SC– 10D, Hokanson, Inc. Bellevue, WA, USA) was wrapped as proximally as possible on the right thigh. This cuff was controlled via a rapid-inflation system (Hokanson E20, Hokanson, Inc. Bellevue, WA, USA) to produce arterial occlusions using 250 mmHg of pressure.

Prior to testing, participants were transferred from their wheelchair to a mat via a ceiling lift. Participants then lay in a supine position for at least 10 min while the testing site was prepared by removing excess hair and cleaning the skin with alcohol pads. Each participant was also provided with a pillow for head and neck support, and if necessary, an additional popliteal bolster was provided to stabilize the knee joint. Each participant was also offered an optional pair of noise dampening headphones to wear during testing. Finally, the lower extremity was stabilized during testing using manual support.

Testing began with an initial 30-s cuff occlusion which was used to assess signal quality. Following a 3–5-min refractory period, an ischemic calibration was completed to establish maximum and minimum HHb values using a 10-s period of surface neuromuscular electrical stimulation (NMES; Theratouch 4.7; Richmar, Inola, OK, USA; biphasic waveform, 5 Hz, 175 mA, 450 μs) during an arterial occlusion lasting 3–5 min. After another 3–5-min refractory period, a 15-s bout of NMES paired with a cuff occlusion of the same length of time was executed. This was immediately followed by a series of 20 transient arterial occlusions (occlusions 1–5: 5-s on / 5-s off; occlusions 6–10: 5-s on / 10-s off; occlusions 11–20: 10-s on / 20-s off). In cases where spasm occurred during testing, this serial occlusion procedure was repeated and the trial with the least number of artifacts was chosen for data processing.

The NIRS data were collected at 10 Hz using OxySoft (3.0.103.3, Artinis Medical Systems, Elst, Netherlands) and a manufacturer recommended differential pathlength factor of 4. Data

were then transferred for initial processed in Visual 3D (6.01.22, C-motion, Germantown, MD, USA). Raw HHb signals were first corrected for expected changes in blood volume as previously described [42]. The signal was then scaled to observed values during the ischemic calibration. To identify when cuff occlusions occurred during serial occlusion tests, a thresholding technique was applied to the rectified second derivative of the corrected HHb signal. These timepoints were visually confirmed. Slopes of the linear portions of the corrected HHb signal during each of the 20 arterial occlusions were calculated using linear regression. The magnitude of these slopes, along with the time for each occlusion, were exported to JMP Pro (14.3.0; SAS Institute Inc., Cary, NC, USA) and fit to a monoexponential curve. In many cases, early data points were excluded from the final curve fitting due to clearly de-trended values identified following visual inspection. Additional outliers were identified using studentized residuals ($\geq$ 2.5) and removed from final curve fitting. In the event that > 50% of the data points were excluded or removed, then that participant's data wereexcluded from analysis. The recovery rate constant (k), $s^{-1}$, of the final monoexponential curve was indicative of HHb recovery and mitochondrial capacity [35]. These adapted methods have been reported in detail [43].

## Statistical analysis

Statistical analysis was performed using SPSS 24 (Chicago, IL, USA). Outliers and normality were identified and assessed for each variable using box plots and Q-Q plots, respectively. Independent t-tests were utilized for the study participant demographic comparisons. Pearson's correlation coefficients were used to identify associations between PBMC mitochondrial activity and skeletal muscle biopsy mitochondrial activity, as well as between mitochondrial capacity as measured by NIRS and skeletal muscle biopsy mitochondrial activity. Partial correlations were conducted to adjust for age, weight, BMI, and time since injury (TSI) between PBMCs and permeabilized muscle fibers [9, 44]. These statistical tools however, are not sensitive enough to establish agreement between different measurement techniques; thus, agreement between the compared methods was assessed using Bland-Altman analysis [45–47]. Due to the presence of different units between PBMCs and skeletal muscle respirometry, standardized Z-scores were computed for each measure prior to Bland-Altman analysis [20]. Confidence intervals $\leq$ 2.5 were used to determine Bland-Altman agreement between the compared methods. Bland-Altman plots graphically display the mean differences between two measurements along with their 95% confidence intervals, thereby allowing for a visual assessment of agreement [45–47]. Paired t-tests were also performed to evaluate potential mean biases between the compared methods [46]. The combination of these complementary statistical tools provides a more complete assessment of agreement between the PBMC measurement technique compared to the muscle biopsy gold standard [45–47]. Paired t-tests were performed comparing individual oxygen consumption rates following the addition of 1mM ADP and 10 uM cytochrome *c* to validate mitochondrial membrane integrity in PMBCs and skeletal muscles. Statistical significance levels were set to an $\alpha$ < 0.05.

## Results

### Participant demographics

Twenty-two individuals were recruited for this study. Two participants withdrew prior to any testing for unknown personal reasons. Demographic and injury information for the remaining 20 participants are summarized in Table 1. No significant differences for age, height, weight, BMI, or TSI were observed between paraplegics (level of injury [LOI]: T1 –S5) and tetraplegics (LOI: C5 –T1), as well as between Caucasians and African Americans.

**Table 1. Participant demographics.**

|  | Tetraplegic | Paraplegic | Total |
|---|---|---|---|
| N | 7 | 13 | 20 |
| AIS, n | A = 4; B = 2; C = 1 | A = 8; B = 1; C = 4 | – |
| Age, year | 39.6 ± 11.4 | 36.6 ± 12.2 | 38 ± 13.0 |
| Weight, kg | 61 ± 11.7 | 72.4 ± 17.0 | 68 ± 16 |
| Height, cm | 178.5 ± 7.8 | 173.1 ± 9.1 | 175 ± 9.0 |
| LOI | C5 –C7 | T1 –T11 | – |
| BMI, kg/m2 | 19.4 ± 5.0 | 24.2 ± 5.5 | 23.0 ± 5.0 |
| TSI, year | 13.7 ± 13.9 | 4.8 ± 3.1 | 8.0 ± 11.0 |
| Caucasian, n | 3 | 6 | 9 |
| African-American, n | 4 | 7 | 11 |
| Male | 6 | 10 | 16 |
| Female | 1 | 3 | 4 |

n, number of participants; AIS, American spinal injury association impairment scale; LOI, level of injury; TSI, times since injury; mean ± SD.

## Relationships between PBMCs and permeabilized skeletal muscle fibers

While PBMC respirometry analysis was performed on all 20 participants, due to insufficient muscle tissue in one sample (*i.e.* wet muscle weight less than 1 mg), permeabilized skeletal muscle respirometry analysis was conducted for 19 participants. Subsequent testing error and equipment malfunction similar to failure of the oxygen sensors in the Oxygraph-2k chamber resulted in the loss of all permeabilized skeletal muscle data for an additional 2 participants and the loss of 1 data point for PBMC complex I and IV. Initial analysis also revealed 3 data points among PBMC complex II measures, 2 data points among PBMC complex IV, and 1 data point among permeabilized skeletal muscle complex II and complex IV to be outliers using Q-Q plots. Normality was found among each of the variables. Table 2 provides mean values of mitochondrial respiration measured from PBMCs and permeabilized skeletal muscle

**Table 2. Bioenergetic profiles of SCI study subjects.**

| Tissue | Complex | Mean | SD | Range |
|---|---|---|---|---|
| Permeabilized muscle fiber | I[&] (n = 17) | 11.22 | 6.10 | 3.06–22.29 |
|  | II (n = 16) | 8.09 | 2.51 | 3.97–12.65 |
|  | II[#] (n = 17) | 8.66 | 3.39 | 3.97–17.79 |
|  | IV (n = 16) | 54.06 | 30.67 | 22.71–118.53 |
|  | IV[#] (n = 17) | 61.23 | 41.92 | 22.71–176.03 |
| PBMC | I (n = 19) | 8.08 | 3.57 | 1.74–14.11 |
|  | I[#] (n = 20) | 9.72 | 8.09 | 1.74–40.74 |
|  | II (n = 17) | 4.08 | 2.61 | 0.09–8.68 |
|  | II[#] (n = 20) | 6.67 | 7.13 | 0.09–29.28 |
|  | IV (n = 17) | 39.18 | 18.60 | 17.30–74.32 |
|  | IV[#] (n = 19) | 47.88 | 33.90 | 17.30–160.01 |
| NIRS (n = 12) | – | 0.011 | 0.004 | 0.004–0.018 |

Respiration is measured as $pmol \cdot sec^{-1} \cdot 1 \times 10^6 \ cells^{-1}$ [PBMCs], $pmol \cdot sec^{-1} \cdot mg \ wet \ weight^{-1}$ [permeabilized fibers], and $s^{-1}$ [NIRS]. $1 \ nmol \cdot min^{-1} = 1000 \ pmol \cdot 60 sec^{-1}$.

PBMC, peripheral blood mononuclear cells; NIRS, near-infrared spectroscopy; SD, standard deviation

[&], data present with outliers excluded

[#], data with outliers included.

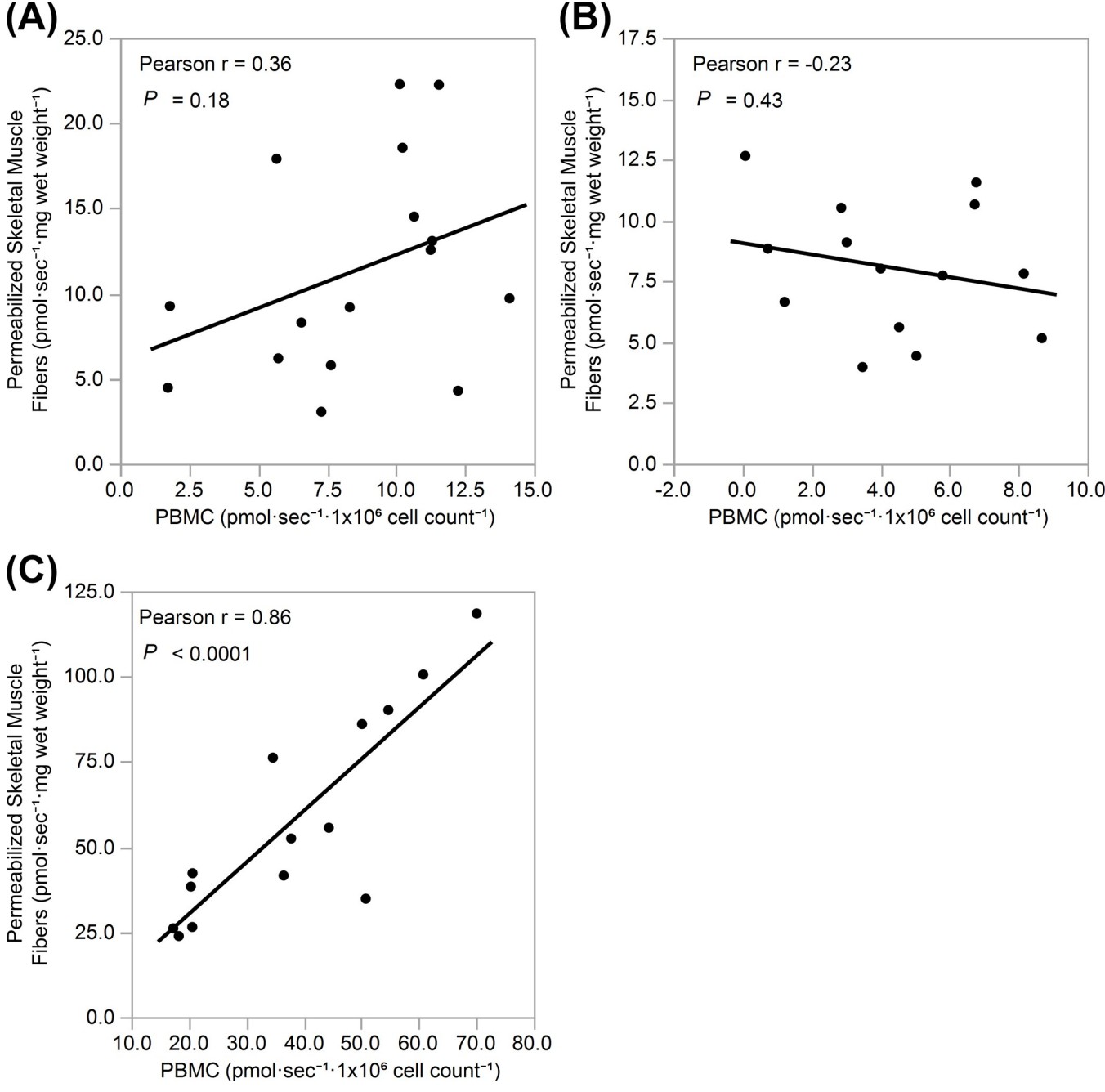

**Fig 2. Comparison of PBMCs and permeabilized skeletal muscle fiber measures of mitochondrial capacity.** Relationships of maximal rate of oxidative phosphorylation between PBMCs and permeabilized muscle fibers for mitochondrial complexes **(A)** I (n = 17), **(B)** II (n = 14), and **(C)** IV (n = 14).

fibers. Comparisons of 16 participants were made for complex I, while comparisons among 14 participants were made for both complex II and IV. Pearson's correlations were used to compare bioenergetics between PBMCs and skeletal muscle fibers (Fig 2). A significant positive relationship was observed between maximal oxidative phosphorylation rates of PBMCs and permeabilized skeletal muscle for complex IV (r = 0.86, P < 0.0001). Conversely, no significant relationships were observed for complex I (r = 0.36, P = 0.18) and complex II (r = -0.23,

**Table 3. Partial correlations between permeabilized muscle fiber and monocyte or NIRS bioenergetics after independently controlling for age, weight, BMI, or TSI.**

| Confounding variables | Complex | PBMCs | | | NIRS | | |
|---|---|---|---|---|---|---|---|
| | | n | Pearson's *r* | *P*-value | n | Pearson's *r* | *P*-value |
| Age | I | 17 | 0.275 | 0.302 | 9 | 0.504 | 0.202 |
| | II | 14 | -0.274 | 0.366 | 8 | -0.471 | 0.286 |
| | IV | 14 | 0.895 | <0.0001* | 9 | -0.585 | 0.127 |
| Weight | I | 17 | 0.307 | 0.248 | 9 | 0.453 | 0.260 |
| | II | 14 | -0.227 | 0.455 | 8 | -0.270 | 0.558 |
| | IV | 14 | 0.900 | <0.0001* | 9 | -0.477 | 0.232 |
| BMI | I | 17 | 0.296 | 0.265 | 9 | 0.335 | 0.418 |
| | II | 14 | -0.228 | 0.454 | 8 | -0.232 | 0.616 |
| | IV | 14 | 0.875 | <0.0001* | 9 | -0.467 | 0.243 |
| TSI | I | 17 | 0.299 | 0.260 | 9 | 0.436 | 0.280 |
| | II | 14 | -0.095 | 0.757 | 8 | -0.583 | 0.169 |
| | IV | 14 | 0.851 | <0.0001* | 9 | -0.514 | 0.192 |

"*" *P*-value ≤ 0.05.

*P* = 0.43). Paired t-tests performed to validate mitochondria membrane integrity yielded no significant difference between individual oxygen consumption rates after adding 1mM ADP and 10 μM cytochrome c in skeletal muscle (*P* = 0.72) and PMBCs (*P* = 0.08).

Partial correlations adjusting for participants' physical characteristics (age, weight, BMI, and TSI) did not yield additional significant relationships (Table 3). Statistically significant positive relationships between PBMCs and permeabilized muscle fibers were maintained when controlling for all physical characteristics. Bland-Altman plots comparing standardized Z-scores of PBMCs and permeabilized fibers (Fig 3) show good agreement for complex I (95% CI: 2.26) and IV (95% CI: 1.03) with no significant biases (*P* > 0.05). The confidence interval for complex II Z-score comparisons fell outside the *a priori* limits of agreement (95% CI: 3.23). Additional details regarding Bland-Altman agreement between PBMCs and permeabilized muscle fibers are provided in Table 4.

### Relationships between NIRS and permeabilized skeletal muscle fibers

Prior to NIRS collection, 2 participants withdrew due to reported pain/sensitivity evoked by the electrical stimulation parameters when combined with cuff occlusions. Due to technical issues, 6 more participants were unable to provide usable NIRS data [43]. This included 4 individuals whose skin adipose tissue exceeded the penetrating depth of the NIRS device employed, and 2 participants' whose data were lost due to quality issues encountered either during collection (*e.g.* muscle spasms) or in processing. The remaining data were found to be free from outliers and normally distributed.

The NIRS recovery profiles for the remaining 12 participants are detailed in Table 2. Comparisons of 9 participants were made for both complex I and IV, while comparisons among 8 participants were made for complex II. No significant correlations were observed between the NIRS recovery rates and the permeabilized skeletal muscle fiber oxidative phosphorylation rates for complex 1 (r = 0.47, *P* = 0.20), complex II (r = -0.22, *P* = 0.60, or complex IV (r = -0.37, *P* = 0.32); additionally, no significant changes were noted when controlling for age, weight, BMI, or TSI (Table 3). Bland-Altman analyses comparing NIRS and permeabilized skeletal muscle fibers were not conducted due to the limited NIRS sample size.

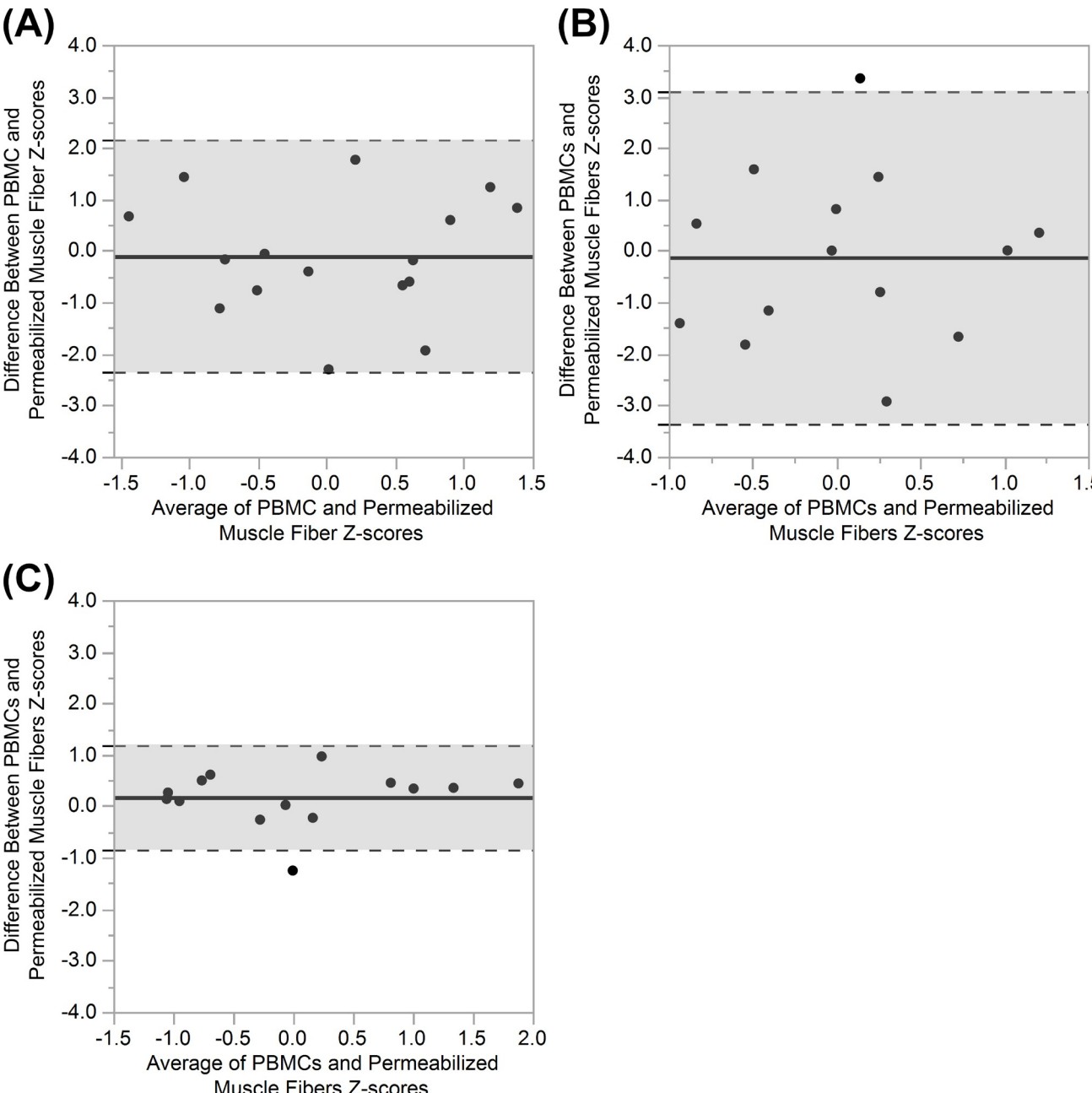

**Fig 3. Agreement between PBMCs and permeabilized skeletal muscle fiber measures of mitochondrial capacity.** Bland-Altman plots of normalized high resolution repirometry of PBMC and normalized permeabilized muscle fiber indices of mitochondrial respiratory capacity in individuals with SCI for mitochondrial complexes **(A)** I (n = 16), **(B)** II (n = 14), and **(C)** IV (n = 14). Solid black line represents the mean difference; the greyed region between the dashed black lines represents the 95% limits of agreement.

## Discussion

This study offers evidence that mitochondrial respiratory capacity as measured with PBMCs may be comparable to the gold standard method utilizing skeletal muscle fibers for persons with SCI. Measurement with NIRS, however, could not be compared with the individual mitochondrial complexes in skeletal muscle fibers and thus its utilization potential in this clinical

**Table 4. Bland-Altman limits of agreement analysis between permeabilized muscle fibers and monocytes Z-scores.**

| | Mean difference (SD) | 95% limits of agreement | |
|---|---|---|---|
| | | Lower limit (95% CI) | Upper limit (95% CI) |
| **PBMCs** | | | |
| Complex I (n = 16) | -0.102 (1.154) | -2.364 (-3.414 to -1.313) | 2.159 (1.108 to 3.210) |
| Complex II (n = 14) | -0.121 (1.648) | -3.350 (-4.976 to -1.724) | 3.108 (1.482 to 4.734) |
| Complex IV (n = 14) | 0.175 (0.523) | -0.851 (-1.367 to -0.334) | 1.200 (0.684 to 1.717) |

SD, standard deviation; CI, confidence interval.

population could not be confirmed. The current study was designed to examine the feasibility and appropriateness of using less and/or noninvasive technologies when assessing mitochondrial capacity following SCI; as such, only comparisons against established permeabilized muscle fiber techniques were considered. The primary findings revealed high-resolution respirometry values for skeletal muscles and PBMCs oxygen consumption rates to be positively related for complex IV independent of age, weight, BMI, and TSI, while also displaying good agreement. Our hypothesis that mitochondrial capacity in persons with SCI could be reliably assessed with alternatively less invasive techniques is therefore only partially supported.

Mitochondria are critical in maintaining traditional skeletal muscle function and vitality. Deficiencies in their function are present in a variety of neurodegenerative and cardiovascular disease states, as a normal component of cellular senescence in aging, as well as metabolic disorders including T2DM and obesity [1, 48–50]. Oxidative capacity of skeletal muscles, reflected by substrate utilization and oxygen uptake of skeletal muscles, typically decrease as a consequence of aging, reduced physical activity, and injury/disease in rodents and humans [36–38, 51–54]. Decreased oxidative capacity, mitochondrial size and protein concentrations have been implicated in individuals with SCI compared to those who were able-bodied [14, 55–57]. While the pathways responsible for mitochondrial dysfunction in individuals with SCI remain unclear, they may be mediated by several mechanisms including oxidative stress [58], apoptosis [59], and aberrant mitophagy [60]. Establishing the utilization of PBMCs and/or NIRS as alternative methods for providing physiological information regarding mitochondrial health for individuals with SCI has yet to be accomplished, thus the findings of this study may serve to provide additional support for this important foundation.

## Mitochondrial respiration in PBMC and permeabilized skeletal muscle fibers

Recently, potential links between mitochondrial health measured in white blood cells with various disease states and disorders have been explored, highlighting their potential as a reliable and convenient source for assessing mitochondrial bioenergetics in translational research [19, 24, 26, 61–64]. Specifically, PBMCs have been recognized as a potentially sensitive marker for mitochondrial dysfunction in a number of neurodegenerative and pathological disease states [24, 26, 63, 65]. Amongst individual oxygen consumption rates, complex IV in both PBMC and skeletal muscle consistently yielded the highest oxygen consumption rates, followed by complex I and complex II (Table 2). To the best of our knowledge, this is the first study to demonstrate evidence of complex IV in PBMCs as a surrogate for complex IV respiratory rates in skeletal muscle; additionally, no previous assessments of PBMC respiration rates for complex II and IV have been performed. Mean PBMC respiration rates for complex I (Table 2) were comparable to able-bodied subjects in similar studies [21, 66, 67]. In contrast,

permeabilized skeletal muscle fiber respiration rates (Table 2) were lower among participants in the current study versus able-bodied populations reported in the literature [20, 21]. These opposing observations could be the result of the differences in mitochondrial respirometry protocols and/or instrumentation implemented between studies, as such making comparisons challenging. Alternatively, this may suggest mitochondrial function is inherently lower in the skeletal muscle fibers for individuals with SCI. However, the use of the robust substrate, TMPD-ascorbate (with corresponding inhibitor to provide a background subtracted rate), can nonetheless provide useful estimates of skeletal muscle oxidative function despite a small sample size. Furthermore, confidence in our low measures for complex I and II respiratory rates is supported by the lack of significance observed in the *a posteriori* normalization techniques utilized in the current study.

Attenuation in the function of complex I occurs in individuals with several disease states including T2DM [68] and Huntington's disease [66], as well as ischemia damaged murine cardiac mitochondria [69]. To identify the potential defect in complex I activity in persons with SCI, respiratory enzyme activities of the electron transport chain need to be studied in future work. Respiration rates using combined complex I+II substrates measured in prior studies [19, 20], was not assessed in the current study. In the present study, separate substrate and inhibitors were utilized to localize defects in the electron transport chain. The assessment of complex I+II respiration rates should be considered in a future study to provide a more physiological relevant assessment of maximal mitochondrial respiratory capacity. However, the measure of complex IV rates may provide a downstream rate similar to complex I+II; thus, providing a greater signal to noise ratio in both skeletal muscle and PBMCs.

The general lack of correlation and Bland-Altman agreements for complexes I and II contrasts that of a study showing strong correlation of complex I in a study evaluating peripheral monocytes and permeabilized skeletal muscle mitochondrial function in healthy non-human primates [19], suggesting a potential impact of SCI on complex I function. In prior studies involving elderly adults, respirometry profiles of PBMCs have been shown to be positively associated with markers of physical function and strength; however, comparison between individual mitochondrial complexes were not evaluated [24, 26]. Partial correlations adjusting for age, weight, BMI, and TSI were performed in the current study primarily due to the wide range in the demographic variables. Large demographic ranges were also present in previous work when comparing mitochondrial mass and activity after SCI, trends and significant changes in activities remained consistent independent of demographic factors including age, TSI, and level of injury [9]. The fact that significant positive correlations remained after independently controlling for age, weight, BMI, and TSI, provide additional support for the correlations observed.

A study investigating the adaptation of 2-week high-intensity interval training in ten young able-bodied men recently demonstrated that PBMCs do not reflect mitochondrial function of skeletal muscles [21]. Although the results of the current study involving complex I is consistent, the utilization of different study populations in each investigation should be carefully noted. Additionally, Hedges *et al.* did not examine the individual respiration rates for complex II and complex IV. Following SCI, several profound physiological changes are observed, including long-term negative effects on mitochondrial regulation and activity [70]. While, Hedges *et al.* enrolled homogenous, able-bodied young male participants, the current study successfully recruited a heterogenous group of individuals with chronic SCI. Absolute maximal oxygen uptake ($VO_2$) peaks of the healthy participants were significantly (2.1–3.6-fold) higher than those typically observed in SCI populations [71, 72]. Unlike for skeletal muscle, the physiologic mechanisms for whether mitochondrial respiratory capacity in less-oxidative tissue (*e.g.* PBMCs) should increase in response to physical training is less clear. The

mitochondrial capacity of PBMCs appear to be less responsive to change in the direction of an increase in respiratory capacity following acute high-intensity training [21], but may be more susceptible to become impaired in chronic diseases states that consist of systemic factors including inflammation, oxidative stress, and/or metabolic syndrome that are evident in individuals with SCI [70, 73, 74]. Therefore, PBMCs may still serve as a useful biomarker of muscle mitochondrial function in individuals with SCI, a pathological population with chronically compromised mitochondrial respiratory capacity compared to their able-bodied counterparts. This is further supported by a recent study revealing that telomeres in PBMCs, a senescence biomarker, from sarcopenic older individuals were shorter relative to non-sarcopenic peers [75]. Since telomeric shortening in PBMCs appears to be reflective of muscle loss, it is possible that chronic inflammatory and metabolic changes influence mitochondrial function within PBMCs [75, 76]. Overall, compared to those who are able-bodied, individuals with SCI have inherent systematic factors that may negatively influence the detection of mitochondrial respiratory capacity. This may provide a potential explanation for the lack of correlation and agreement for mitochondrial complexes I and II in the current cross-sectional study with untrained persons with SCI. However, the robust positive correlations and agreement observed after the addition of TMPD-ascorbate in PMBCs and skeletal muscles in the current study supports that mitochondrial complex IV function in PBMCs is a useful indicator of mitochondrial function in skeletal muscle of persons with SCI. The minimally invasive procedure of obtaining a blood sample to measure mitochondrial function via PBMCs will allow more research groups and clinicians alike to conduct this type of analysis and facilitate maximum benefit of interventions for rehabilitation. Future work investigating the effects of endurance and resistance training on mitochondrial functional improvement over time for persons with SCI is fundamentally necessary to assess the potential to reduce or limit long-term systemic complications. Having convenient access to mitochondrial health will allow providers to adjust type, duration and intensity of exercise intervention in order to maximize health benefits.

## Mitochondrial respiration in skeletal muscle fiber measured in NIRS and OXPHOS

The average NIRS recovery rate measured in the current study (0.011 $s^{-1}$) was consistent with literature reported average values among other SCI groups ranging from 0.008 to 0.017 $s^{-1}$ [12, 36, 77, 78]. The average recovery rate observed in this investigation fell below previously reported able-bodied cohorts by greater than 60% [20, 36, 37, 77, 79, 80]. This observation is consistent with prior work performed by Erickson *et al.* [36] comparing the SCI group with able-bodied controls.

Previous findings highly suggested that NIRS is an effective tool to measure overall mitochondrial oxidative capacity. However, the current findings did not support the hypothesis that NIRS could be used to measure mitochondrial complexes similar to what has been reported in the able-bodied population [20]. A lack of significant correlations alone does not imply poor agreement between the NIRS and permeabilized skeletal muscle fiber results as these measures may be somewhat misleading, especially when considering our small heterogeneous sample [45–47]. However, using NIRS to measure oxidative capacity in persons with SCI has proven to be more technically challenging than in able-bodied groups [43]. These challenges resulted in a diminished sample size and the exclusion of Bland-Altman analyses comparing NIRS and permeabilized skeletal muscle fiber techniques. Previous comparisons of these tools in an able-bodied population by Ryan *et al.* [20] displayed good agreement for complex I (95% CI: 2.43) and complex II (95% CI: 1.53) while complex IV was not explored. Researchers should consider examining the agreement between mitochondrial supercomplex

activity and NIRS, as the nature of this noninvasive tool may be better suited to capture global oxidative processes.

## Limitations

The relatively small sample sizes in the current study increases the likelihood for type II error and limits the strength of correlation. The small sample of this very heterogeneous population with multiple factors influencing their physiologic health could also explain the presence of the outliers in the analysis. With outliers included, however, the mean respiration rates were largely unaffected. The presence of greater and variable amounts of intramuscular fat among the biopsy samples [7], in addition to not correcting for auto-oxidation (for both muscle and PBMCs), may have influenced the respiration rates, further complicating the analysis and potentially explaining some of our outliers. However, using inhibitor sensitive rates in the current study was deemed a better alternative approach since rates are more specific than correcting for auto-oxidation [41, 81]. Maximal respiratory capacity, an important parameter for determining mitochondrial dysfunction, was not assessed due to limited muscle biopsy size and quality, and thus should be considered in future work. Unfortunately, a number of methodological issues arose using NIRS, dramatically reducing the sample size and limiting the utilization of Bland-Altman comparisons. Furthermore, capturing repeated NIRS data across multiple days or from the opposite limb may have allowed for establishing test-retest reliability measures; however, due to difficulties with transportation to-and-from the medical center, the length of time necessary for testing, and potential interlimb variability these additional steps were not considered. Including an able-bodied matched cohort would also provide additional strength to the study; however, the design of the current work did not require such an addition as our aim was to compare techniques to assess mitochondrial health in SCI. At the expense of facilitating easily interpretable results, it was necessary to reconcile unit differences between comparisons using Z-score; future studies should explore alternative tools that may not be limited in this fashion. While current high-resolution respirometry techniques employed in this study require small volumes of blood to assess OXPHOS rates, PBMCs naturally contain a lower density of mitochondria when compared to skeletal muscle [65]. Such differences may artificially influence bioenergetic outcomes, resulting in less accurate assessments of mitochondrial capacity, particularly for complex I and II in the current study. Additionally, larger or more homogenous samples may allow researchers to accurately estimate the minimal detectible change values for these noninvasive assessment techniques. Due to limitations in biopsy and blood samples collected, reliability data were not performed in the current study, however, will be considered in future work.

## Conclusion

This preliminary study, to the authors' knowledge, is the first to explore and support the agreement of less invasive clinical strategies for assessing mitochondrial respiratory capacity in persons with SCI using PBMCs, particularly for complex IV. Further studies are warranted to provide a measurement of complex I and II activity using less and noninvasive techniques. At this time, NIRS appears to be challenging to implement for investigating individual mitochondrial complex activities in persons with SCI. Overall, the findings may assist in supporting the implementation of PBMCs as an alternative when assessing mitochondrial health in future SCI clinical trials, especially those evaluating exercise interventions. The mitochondrial health information provided through these less-invasive procedures may ultimately serve as a critical marker for clinical decision-making aiding in driving down rates of the many chronic metabolic comorbidities that typically plague individuals with SCI.

## Acknowledgments

The authors would like to acknowledge all our study participants, Laura O'Brien, Ph.D. for establishing funding support, the assistance of Refka Khalil, D.C. for research coordination, Timothy Lavis, M.D., Lance Goetz, M.D., and Teodoro Castillo, M.D. for their help with screenings and physical examinations, Jeremy Thompson, B.S. and Satinder Gill, Ph.D. for technical assistance, and the Hunter Holmes McGuire Department of Veterans Affairs Medical Center for the opportunity to conduct clinical research.

## Author Contributions

**Conceptualization:** Edward J. Lesnefsky, Ashraf S. Gorgey.

**Data curation:** Raymond E. Lai, Matthew E. Holman, Qun Chen, Jeannie Rivers.

**Formal analysis:** Raymond E. Lai, Matthew E. Holman, Qun Chen, Edward J. Lesnefsky, Ashraf S. Gorgey.

**Funding acquisition:** Ashraf S. Gorgey.

**Methodology:** Raymond E. Lai.

**Resources:** Ashraf S. Gorgey.

**Software:** Qun Chen, Edward J. Lesnefsky.

**Supervision:** Edward J. Lesnefsky, Ashraf S. Gorgey.

**Writing – original draft:** Raymond E. Lai, Matthew E. Holman, Ashraf S. Gorgey.

**Writing – review & editing:** Raymond E. Lai, Matthew E. Holman, Edward J. Lesnefsky, Ashraf S. Gorgey.

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
