## [Decision Letter · Decision Letter 0]

19 Oct 2021

PONE-D-21-26520Assessment of mitochondrial respiratory capacity using minimally invasive and noninvasive techniques in persons with spinal cord injuryPLOS ONE

Dear Dr. Gorgey,

Thank you for submitting your manuscript to PLOS ONE. After careful consideration, we feel that it has merit but does not fully meet PLOS ONE’s publication criteria as it currently stands. Therefore, we invite you to submit a revised version of the manuscript that addresses the points raised during the review process.

We look forward to receiving your revised manuscript.

Kind regards,

Todd A. Astorino, Ph.D FACSM

Academic Editor

PLOS ONE

Journal Requirements:

"The authors would like to acknowledge all our study participants, Laura O’Brien, Ph.D. for establishing funding support (ID:02265), the assistance of Refka Khalil, D.C. for research coordination, Timothy Lavis, M.D.,  Lance Goetz, M.D., and Teodoro Castillo, M.D. for their help with screenings and physical examinations, Jeremy Thompson, B.S. and Satinder Gill, Ph.D. for technical assistance, and the Hunter Holmes McGuire Department of Veterans Affairs Medical Center for the opportunity to conduct clinical research"

"AG: Ashraf Gorgey

This study was supported by the DoD-CDRMP (W81XWH-14-SCIRP-CTA).

The funders had no role in study design, data collection and analysis, decision to publish, or preparation of the manuscript"

Additional Editor Comments (if provided):

I appreciated reading your submission and feel that the Reviewers have provided a substantive and fair assessment of your work; that said, I have additional concerns that I would like you all to address in your rebuttal.

1. As muscle samples were acquired from these patients, is there a reason why citrate synthase activity, a well known surrogate of mitochondrial content, was not determined? Or cytochrome oxidase as a surrogate of ETC function? You know that these are widely assayed in metabolic studies performed in people without SCI. I know that these tell the scientist something different than respiratory capacity, but they would likely be associated with some of the outcomes (PBMCs, etc.) presented in this paper.

2. The methods section would benefit from inclusion of a brief Experimental Design section, specifically including text concerning fed state of participants, if they abstained from physical activity for some time before testing, voided their bladder, time of day of testing, etc.

3. Do you have any reliability data for your measures? If so, these would be useful to include in the Methods text.

4. Table 2 seems to suggest that PBMC underestimates many of these outcomes vs. muscle fibers; do you have any explanation for this finding? And is there a reason why paired t-tests were not used to compare these mean values statistically rather than only using Pearson r?

5. Lines 436-439 seem to need a sentence or 2 specifically referring to mitochondrial dysfunction-abnormality in persons with SCI.

6. Were these people with SCI physically active, and if so, it may be worthwhile to report this as improved mitochondrial function would be expected irrespective of their disability status.

Reviewers' comments:

Reviewer's Responses to Questions

**Comments to the Author**

1. Is the manuscript technically sound, and do the data support the conclusions?

Reviewer #1: Yes

Reviewer #2: Yes

2. Has the statistical analysis been performed appropriately and rigorously? 

Reviewer #1: Yes

Reviewer #2: Yes

3. Have the authors made all data underlying the findings in their manuscript fully available?

Reviewer #1: Yes

Reviewer #2: Yes

4. Is the manuscript presented in an intelligible fashion and written in standard English?

Reviewer #1: Yes

Reviewer #2: Yes

5. Review Comments to the Author

Reviewer #1: The authors analyzed 22 individuals to evaluate whether mitochondrial respiration of PBMCs and NIRS are predictive of respiration of permeabilized muscle fibers after SCI. The results showed positive correlation between PBMC and permeabilized muscles fibers. However, no association was found between NIRS mitochrondrial capacity and respiration.

1. Line 289. Outliers were excluded based on studentized residuals of 2.5. the threshold of 2.5 seems quite liberal. How many data points were removed? Didn’t these data points provide information or the model was not a good one?

2. Only parametric test (T-test, linear regression and correlation) were implemented. What distribution were the data? Whether they satisfy the model assumptions?

3. Line 323. No significant difference for age,….. Please provide the information about statistical approach implemented for these participant demographics comparison in statistical analysis section.

Reviewer #2: Overview

Mitochondrial dysfunction likely contributes to the etiology of T2DM, obesity, and cardiovascular disease in those with SCI. Mitochondrial function is typically measured from permeabilized muscle fibers attained via muscle biopsy, a technique that is difficult to perform in those with SCI. Therefore, the purpose of this study was to examine the validity of two other potential methods of assessing mitochondrial function, mitochondrial respiration of peripheral blood mononuclear cells (PBMCs) and near-infrared spectroscopy (NIRS). Mitochondrial respiration was measured from permeabilized muscle fibers and PBMCs and mitochondrial capacity was measured by NIRS in 21 individuals with complete or incomplete SCI. A positive, significant correlation was found between PBMC and permeabilized muscle fibers for mitochondrial complex IV, but no relationship was found for NIRS. This study provides some evidence that PBMCs can be used to assess mitochondrial function using a minimally invasive procedure in those with SCI.

Specific comments

Lines 68-70 How does SCI result in mitochondrial dysfunction?

Line 71 It is easy to see why the measurement of mitochondrial function is important to measure to assess patient prognosis, but it would be good to know more about its relevance during initial diagnosis shortly following injury. How would the assessment of mitochondrial function in those with SCI be used by clinicians to treat their patients more effectively?

Lines 76-83 It is difficult to follow the relevance of the previous research results mentioned here. It is easy to understand that mitochondrial function decreases with age, but this occurs in those without SCI as well. The effects of mitochondrial function on substrate use are certainly applicable here, but there is a plethora of evidence that heavy reliance on carbohydrate occurs in those with SCI during voluntary exercise, a finding with that is more generalizable than acute electrical stimulation. In this section, it would be good to know more precisely how mitochondrial dysfunction in those with SCI leads to an increased risk for T2DM, obesity and cardiovascular disease.

Line 244 When were the NIRS measurements taken relative to the other measures and were subjects tested in an overnight fasted state?

Line 406 “Data” is plural. Thus, follow it with “were” rather than “was” here and throughout.

Line 419 While the authors adequately explain the likely mechanisms for their primary findings, they do little to address the applicability of the data. Under what context might a clinician use a measurement of mitochondrial respiration of PBMCs to alter the treatment of their patients? It is interesting that a minimally invasive technique may adequately estimate mitochondrial function in those with SCI, but these results have little meaning if the reader cannot see how they would actually be applied to improve the lives of those with SCI.

6. PLOS authors have the option to publish the peer review history of their article (what does this mean?). If published, this will include your full peer review and any attached files.

Reviewer #1: No

Reviewer #2: **Yes: **Kevin Jacobs

---

## [Author Response · Author response to Decision Letter 0]

9 Feb 2022

Additional Editor Comments:

Q1. As muscle samples were acquired from these patients, is there a reason why citrate synthase activity, a well known surrogate of mitochondrial content, was not determined? Or cytochrome oxidase as a surrogate of ETC function? You know that these are widely assayed in metabolic studies performed in people without SCI. I know that these tell the scientist something different than respiratory capacity, but they would likely be associated with some of the outcomes (PBMCs, etc.) presented in this paper.

Response: Thank you for your comment and suggestion. We absolutely agree that citrate synthase serves as a surrogate for mitochondrial content. The activity of citrate synthase (CS) is measured by mitochondrial enzyme activity assays which is beyond the scope of our research question in the current manuscript. Mitochondrial enzyme assays performed analyzing skeletal muscle biopsy samples is actually being explored in other work we have in the pipeline where we are measuring CS activity as a surrogate for mitochondrial mass. The focus of our current submission is non-invasive strategies to assess mitochondrial function. From our understanding, we are unaware of how to obtain and measure CS content though PBMCs and NIRS testing. 

Q2. The methods section would benefit from inclusion of a brief Experimental Design section, specifically including text concerning fed state of participants, if they abstained from physical activity for some time before testing, voided their bladder, time of day of testing, etc.

Response: Participants were enrolled in the trial as a part of an ongoing clinical trial looking at the effect of 2-different paradigms of electrical stimulation. The current study was conducted with participant data at the baseline before being enrolled in any intervention. Participants fasted for 10-12 hours and abstained from any level of physical activities 2-3 days prior to the intervention; additionally, our participants were motor complete injury and characterized by a very low-level of physical activity [Line 152]. After measuring basal metabolic rate (6.00-6.30 AM), blood was collected for measuring biomarkers, insulin sensitivity and glucose tolerance. Participants were then admitted to the minor procedure room around 12.00-1.00 to conduct muscle biopsy as explained in the trial. We’ve added a “study design” sub-section in the methods to provide additional clarity [Lines 164-170]. 

Q3. Do you have any reliability data for your measures? If so, these would be useful to include in the Methods text.

Response: We would like to thank the reviewer for this comment. Because of IRB constraints and the acquired sarcopenia in persons with SCI, we were only limited to small muscle tissue as well as the amount of blood work necessary to conduct the trial. Additionally, the NIRS testing and set up process may take additional 2-3 hours to conduct particularly when adjusting to the necessary settings, which is considered labor-intense and requires additional time commitment from our subjects. However, we are fully aware of the significance of conducting future reliability study on these measures and we have acknowledged this point as one of the limitations of the study [Lines 595-596]. 

Q4. Table 2 seems to suggest that PBMC underestimates many of these outcomes vs. muscle fibers; do you have any explanation for this finding? And is there a reason why paired t-tests were not used to compare these mean values statistically rather than only using Pearson r?

Response: 

We believe that the amount of the cells in the drawn blood, especially in persons with SCI, is lower and may require increased volume to reflect the true mitochondrial respiration. Additionally, persons with SCI characterized by systematic inflammation that is likely to impair PBMC respiration. This point was stated in the discussion section [Lines 521-522 and Lines 523-525]. Previous trials that addressed similar research questions (Tyrell et al, 2016 Ref#19, Hedges, et al 2019 Ref #21, Ryan et al.2014, Ref #20) have used both Pearson r and Bland-Altman analysis. 

Paired t-test analysis was performed and is detailed in the methods section [Lines 327-329] as well as in the results section [Line 383]. No significant biases were measured using the paired t-tests. Comparisons made using paired t-tests to assess for biases between each of the normalized complexes revealed no significant differences (I: mean difference: -0.10, P = 0.73; II: mean difference: -0.12, P = 0.79; I: mean difference: 0.17, P = 0.23). 

Q5. Lines 436-439 seem to need a sentence or 2 specifically referring to mitochondrial dysfunction-abnormality in persons with SCI.

Response: Thank you for your comment. We have added a sentence to clarify this point [Lines 436-438]

Q6. Were these people with SCI physically active, and if so, it may be worthwhile to report this as improved mitochondrial function would be expected irrespective of their disability status.

Response: The subjects with SCI were not physically active. All of our participants were motor complete and were characterized by very low level of physical activity [Lines 155-158]. We have incorporated this in the methods section [Line 152]. 

Reviewer #1: The authors analyzed 22 individuals to evaluate whether mitochondrial respiration of PBMCs and NIRS are predictive of respiration of permeabilized muscle fibers after SCI. The results showed positive correlation between PBMC and permeabilized muscles fibers. However, no association was found between NIRS mitochondrial capacity and respiration.

Q1. Line 289. Outliers were excluded based on studentized residuals of 2.5. the threshold of 2.5 seems quite liberal. How many data points were removed? Didn’t these data points provide information or the model was not a good one?

Response: In an attempt to retain as much of the original data as possible, we elected to utilize a threshold of 2.5 which encapsulates over 98% of all normalized data. The effects and number of these outliers were varied across individuals and time points (as detailed in the cited paper Ref#43: Ghatas MP et al., 2019), but generally did not allow for a good fit of the normal physiologic monoexponential curves. For a detailed review and justification of this methodological approach please see the aforementioned publication.

Q2. Only parametric test (T-test, linear regression and correlation) were implemented. What distribution were the data? Whether they satisfy the model assumptions?

Response: Our data were all normally distributed prior to analysis as determined using Q-Q plots with outliers removed first as determined using box plot analysis. Mean differences calculated for paired t-tests were all normally distributed as determined using Q-Q plots) [Lines 358-359]. All other paired t-test assumptions were met. Analysis of residual plots showed constant variance among our correlation analyses. All other correlation assumptions were met. 

Q3. Line 323. No significant difference for age,….. Please provide the information about statistical approach implemented for these participant demographics comparison in statistical analysis section.

Response: Independent t-tests were performed. We have added this to the statistical analysis section [Lines 314-315]. 

Reviewer #2: Overview

Mitochondrial dysfunction likely contributes to the etiology of T2DM, obesity, and cardiovascular disease in those with SCI. Mitochondrial function is typically measured from permeabilized muscle fibers attained via muscle biopsy, a technique that is difficult to perform in those with SCI. Therefore, the purpose of this study was to examine the validity of two other potential methods of assessing mitochondrial function, mitochondrial respiration of peripheral blood mononuclear cells (PBMCs) and near-infrared spectroscopy (NIRS). Mitochondrial respiration was measured from permeabilized muscle fibers and PBMCs and mitochondrial capacity was measured by NIRS in 21 individuals with complete or incomplete SCI. A positive, significant correlation was found between PBMC and permeabilized muscle fibers for mitochondrial complex IV, but no relationship was found for NIRS. This study provides some evidence that PBMCs can be used to assess mitochondrial function using a minimally invasive procedure in those with SCI.

Specific comments

Q1. Lines 68-70 How does SCI result in mitochondrial dysfunction?

Response: Exploring the effects of chronic SCI on mitochondrial function is essentially the question we are trying to uncover through our current work. We have updated these lines in order to clarify that mitochondrial dysfunction may be related with having chronic SCI [Lines 68-70]. We have recently published two reviews [Gorgey A, 2018 (Ref# 70) and Goldsmith et al., 2021 (PMID:33891156)] that highlighted how SCI impacts mitochondrial health. Based on the initial observation (O’Brien et al. 2017 (Ref#44); Sumrell et al. 2018 (PMID: 30169541), we have noticed that increasing visceral adipose tissue (VAT) is associated with mitochondrial dysfunction. We further noticed that increasing VAT is accompanied with increasing biomarkers of inflammation (TNF-alpha). We have then hypothesized that increasing VAT may have lead to increasing circulating inflammation that causes mitochondrial dysfunction (Goldsmith et al. currently under review). Additionally, we cannot rule out the effects of ROS on mitochondrial function and health; which may be mediated by increasing systematic inflammation. 

Q2. Line 71 It is easy to see why the measurement of mitochondrial function is important to measure to assess patient prognosis, but it would be good to know more about its relevance during initial diagnosis shortly following injury. How would the assessment of mitochondrial function in those with SCI be used by clinicians to treat their patients more effectively?

Response: Mitochondrial evaluation has been implemented in many neurologic and cardiovascular conditions. Evaluating mitochondrial function may provide integrative measurements of oxidative capacity which include physiological temperatures, endogenous oxygen delivery systems, and preservation of the mitochondrial reticulum (Willingham et al 2017 Ref#15). The lower cost and portability of NIRS makes the methodology more readily available to researchers and healthcare professionals and increases the potential for integration into clinical practice. These tools may ultimately conveniently guide clinicians to develop and implement strategies to enhance mitochondrial oxidation capacities similar to electrically evoked resistance exercise or delivering pharmaceutical intervention (i.e. testosterone, anti-oxidant medications or MitoQ). Obtaining a better understanding of the cellular response of skeletal muscles in chronic SCI may help clinicians better formulate a therapeutic and/or rehabilitative regimen for improving the long-term health and quality of life for individuals with SCI (Gorgey et al, 2021 Ref #13, Gorgey et al., 2020 Ref#16). We’ve added a few lines in the introduction, discussion, and conclusion to address this point [Lines 83-90, 530-538, 607-610]. Lines 530-538 in the discussion section, suggests that mitochondrial health data in potentially reducing long-term systemic complications. 

Q3. Lines 76-83 It is difficult to follow the relevance of the previous research results mentioned here. It is easy to understand that mitochondrial function decreases with age, but this occurs in those without SCI as well. The effects of mitochondrial function on substrate use are certainly applicable here, but there is a plethora of evidence that heavy reliance on carbohydrate occurs in those with SCI during voluntary exercise, a finding with that is more generalizable than acute electrical stimulation. In this section, it would be good to know more precisely how mitochondrial dysfunction in those with SCI leads to an increased risk for T2DM, obesity and cardiovascular disease

Response: The reviewer has brought several excellent points. Persons with SCI are considered an excellent model of premature aging. We have noticed in our earlier work that following the age of 40; there is a dysfunction in both mitochondrial density and activity. Dysfunction in mitochondrial function will lead to decrease fatty acid oxidation and will result in both accumulation of intra and inter-muscular adipose tissue that are likely to cause insulin resistance and then development of type II diabetes. 

We totally agree with the reviewer that persons with SCI relies primarily on carbohydrates and upper extremity voluntary exercise favors carbohydrate utilization. Even with acute bout of NMES in the no-trained muscle, we have noticed reliance on carbohydrate utilization that may highlight mitochondrial dysfunction (Gorgey and Lawrence, 2016 (Ref#10)).

However, when we performed 12 weeks of resistance training, we have noticed that increase in fat utilization at low-intensity FES cycling compared to carbohydrate utilization; suggesting enhancement of mitochondrial machinery at low-intensity exercise in persons with SCI (Gorgey et al. 2021 (Ref# 13)).

Please refer to our review (Gorgey A et al. 2018; EJAP (Ref #70)) that discussed all these important factors in detail. 

Q4. Line 244 When were the NIRS measurements taken relative to the other measures and were subjects tested in an overnight fasted state?

Response: Subjects underwent an overnight fast and NIRS testing was conducted 3-4 days after the muscle biopsy procedure. For example, subjects typically came in on Thursday for the biopsy and NIRS testing was performed on either Monday or Tuesday morning. We have added this information in lines 163-170.

Q5. Line 406 “Data” is plural. Thus, follow it with “were” rather than “was” here and throughout.

Response: Thank you for this suggested correction. We have updated this throughout the manuscript. 

Q6. Line 419 While the authors adequately explain the likely mechanisms for their primary findings, they do little to address the applicability of the data. Under what context might a clinician use a measurement of mitochondrial respiration of PBMCs to alter the treatment of their patients? It is interesting that a minimally invasive technique may adequately estimate mitochondrial function in those with SCI, but these results have little meaning if the reader cannot see how they would actually be applied to improve the lives of those with SCI.

Response: Thank you for your comments. Please see response from Question 2.

---

## [Editor Report · Decision Letter 1]

24 Feb 2022

Assessment of mitochondrial respiratory capacity using minimally invasive and noninvasive techniques in persons with spinal cord injury

PONE-D-21-26520R1

Dear Dr. Gorgey,

We’re pleased to inform you that your manuscript has been judged scientifically suitable for publication and will be formally accepted for publication once it meets all outstanding technical requirements.

Kind regards,

Todd A. Astorino, Ph.D FACSM

Academic Editor

PLOS ONE

Additional Editor Comments (optional):

I thank you Dr. Gorgey for an effective and substantive rebuttal to my comments as well as those of the Reviewers. The paper was improved and in its current form, is of sufficient quality to merit publication in this Journal. Best of luck in your future work and take care.
---

## [Editor Report · Acceptance letter]

2 Mar 2022

PONE-D-21-26520R1 

Assessment of mitochondrial respiratory capacity using minimally invasive and noninvasive techniques in persons with spinal cord injury 

Dear Dr. Gorgey:

I'm pleased to inform you that your manuscript has been deemed suitable for publication in PLOS ONE. Congratulations! Your manuscript is now with our production department. 

Kind regards, 

on behalf of

Dr. Todd A. Astorino 

Academic Editor

PLOS ONE